# False Negative Results in Cervical Cancer Screening—Risks, Reasons and Implications for Clinical Practice and Public Health

**DOI:** 10.3390/diagnostics12061508

**Published:** 2022-06-20

**Authors:** Anna Macios, Andrzej Nowakowski

**Affiliations:** 1Doctoral School of Translational Medicine, Centre of Postgraduate Medical Education, Marymoncka Street 99/103, 01-813 Warsaw, Poland; 2Department of Cancer Prevention, The Maria Sklodowska-Curie National Research Institute of Oncology, Roentgen Street 5, 02-781 Warsaw, Poland

**Keywords:** cervical cancer, screening, false negative results

## Abstract

False negative (FN) results in cervical cancer (CC) screening pose serious risks to women. We present a comprehensive literature review on the risks and reasons of obtaining the FN results of primary CC screening tests and triage methods and discuss their clinical and public health impact and implications. Misinterpretation or true lack of abnormalities on a slide are the reasons of FN results in cytology and p16/Ki-67 dual-staining. For high-risk human papillomavirus (HPV) molecular tests, those include: truly non-HPV-associated tumors, lesions driven by low-risk HPV types, and clearance of HPV genetic material before sampling. Imprecise disease threshold definition lead to FN results in visual inspection with acetic acid. Lesions with a discrete colposcopic appearance are a source of FN in colposcopic procedures. For FAM19A4 and hsa-miR124-2 genes methylation, those may originate from borderline methylation levels. Histological misinterpretation, sampling, and laboratory errors also play a role in all types of CC screening, as well as reproducibility issue, especially in methods based on human-eye evaluation. Primary HPV-based screening combined with high quality-assured immunocytochemical and molecular triage methods seem to be an optimal approach. Colposcopy with histological evaluation remains the gold standard for diagnosis but requires quality protocols and assurance measures.

## 1. Introduction

The fundamental objective of screening is to distinguish healthy persons from those in an early, asymptomatic phase of certain diseases by the use of available, validated, inexpensive, non-invasive, and widely accepted screening tests [1]. As no ideally accurate screening test exists, planning of screening procedures and algorithms is a matter of balancing potential benefits and harms [2]. Both false negative (FN) and false positive (FP) results, i.e., negative results in persons with the disease and positive results in healthy persons, respectively, should be maximally limited to minimize the risk of progression of the disease that escaped detection, unfavorable prognosis, and delayed treatment, as well as overdiagnosis, overtreatment, high costs, and needless anxiety [3].

The aim of this work is to discuss the risks and reasons of FN results of cervical cancer (CC) screening tests and triage methods, as well as their implications on screening process.

### 1.1. Cervical Cancer Neoplasia

A great majority of cervical neoplasia is driven by infections with oncogenic types of human papillomavirus (HPV) [4], and the development of only a small fraction of neoplastic lesions is considered HPV-independent [5]. In most cases, HPV infections acquired mainly by sexual contacts are self-limiting and regress spontaneously; however, a fraction of them progress to cancer precursor lesions, termed cervical intraepithelial neoplasia (CIN), and rarely to invasive cancer [6,7,8]. At each stage of CIN, lesions may regress spontaneously, but the probability of regression decreases when lesion severity increases [9,10]. Currently, mild CIN (CIN1) is considered a benign cellular manifestation of productive HPV infection with high regression potential, which, in most cases, requires only surveillance [11]. Moderate CIN (CIN2) and severe CIN (CIN3), which is the direct precursor of cancer, require treatment due to their higher risk of progression [12]. Ablative or excisional treatment of CIN2/CIN3 is used in non-pregnant women to avoid progression to cancer [13]. The period between HPV infection and cancer invasion takes at least 10–12 years in a great majority of cases [14,15,16]. The natural history of rare, HPV-unrelated cervical neoplasia is not well-established yet.

### 1.2. Cervical Cancer Screening

Due to a long period of precancerous lesions development and the possibility of their accurate identification and effective treatment, CC screening has been introduced in many countries [17,18]. Screening procedures aim at detecting CIN2 and CIN3 lesions in asymptomatic women and providing them appropriate treatment and follow-up. Therefore, screening may reduce CC incidence and, hence, mortality [19]. However, high coverage of the target population and high quality of screening, triage, treatment, and follow-up procedures, as well as the appropriate organization of the screening processes need to be assured to achieve these goals [20].

Exfoliative cervical cytology remains the basic test for CC screening in developed countries. However, as the discovery of the role HPV in the aetiology of CC emerged [21], the possibility of applying new methods based on HPV detection in screening appeared [18,22,23]. HPV-based molecular testing was proven to be approximately 1.5 times more sensitive than cytology-based screening in the detection of CIN2+ and CIN3+ [24]. Low- and middle-income countries (LMIC) rely on cheaper methods, such as visual inspection with acetic acid (VIA), which is recommended by the World Health Organization (WHO) in cases when no other option is available or affordable [25]. Depending on the screening test, different triage methods of screen-positive women are incorporated, including colposcopy with colposcopically-targeted or random biopsies, a mixture of other screening tests, or immediate treatment [20]. Other methods, such as dual staining (DS) for the simultaneous expression of p16 and Ki-67 proteins in cells or detecting hypermethylation of the FAM19A4 and miR124-2 genes, are still being extensively studied and gradually implemented into opportunistic screening programs [26,27]. Nevertheless, none of CC screening and triage tests and methods have perfect accuracy. Limited specificity results in FP results. Less than perfect sensitivity poses a risk of more dangerous FN results and consequences, which are studied and discussed in this paper (Table 1).

## 2. CC Screening and Triage Tests and Methods, Their Performance, and Reasons for False-Negative Results

### 2.1. Screening Tests

#### 2.1.1. Cytology 

For decades, the Pap smear remained a basic screening test for the secondary prevention of CC in the more affluent countries and has been proven to be very effective in the reduction of CC incidence [28] and mortality [29]. Originally, cytology was performed as a conventional smear (Pap test). Over time, liquid-based cytology (LBC) was developed to reduce the fraction of slides unsatisfactory for evaluation obtained while using the Pap test. LBC utilizes a liquid medium to preserve cells, which are subsequently transferred to a slide in a thin layer [20]. The rate of slides that were inadequate for evaluation due to overlapping cells, obscuring blood, or inflammation was reduced by the use of LBC, compared with conventional cytology; however, no overall impact on limiting unsatisfactory evaluation rates was observed [30]. Studies comparing the sensitivity of conventional cytology and LBC provide conflicting results [20] regarding whether LBC is more [31,32] or equally sensitive [33,34,35], compared to Pap test. 

Published studies reported specificity of cytology at 86–100% [36]. However, its limited reproducibility [37] and sensitivity, ranging around 30–87% [36,38], cause a serious risk of missing precancerous lesions. The issue of interval cancers—invasive cancers diagnosed after normal result of screening test within the screening interval [38]—is vital in screening programs that are based on a Pap smear [39,40]. “European guidelines of quality assurance in CC screening” recommended auditing all CC cases and their screening history, with special attention paid to those with negative cytology results prior to cancer diagnosis; all FN slides should be seeded among randomly selected screening slides and rescreened [20]. The audit is, indeed, performed in many countries, with a heterogenous methodology applied [41].

Among the reasons for FN cytological results are: (1) misclassification of an abnormal slide as normal, which results in no referral for further diagnostic examinations; (2) sampling errors; (3) true lack of abnormal cells on a slide due to lesion location—submucosal or deep in the endocervical canal; and (4) rater-dependence, with low reproducibility.

##### Misclassification of Abnormal Slide as a Normal One

Interpretation of a slide as normal, despite the presence of abnormal cells, was pointed out as a reason for FN diagnoses in over 50% of slides in many studies. In a pooled analysis by DeMay, 655 no intraepithelial lesions or malignancy (NILM) slides were evaluated, and 340 of them were reclassified as abnormal (51.9%) [42], which was in line with results obtained by Kenter et al. (53.3%) [43], Bulk et al. (61.1%) [44], and, more recently, Komerska et al. (54.2%) [45]. Other researchers, however, reported the rate of misinterpretation as low as 4.6% to 15% [46,47,48]. Emphasis on certification, training, and proficiency tests for laboratory staff may limit the rate of misclassification [49].

Small number of abnormal cells on the slide was shown to impede proper evaluation [48,50]. In a study by Sherman et al., 50% of re-evaluated false-negative slides had <100 cells with features of squamous intraepithelial lesions [51]. In a similar analysis, 34 out of 43 FN smears (79%) contained <100 abnormal cells, compared to 5/45 in true positive cases (11%) [52].

##### Sampling Errors—Misclassification of Unsatisfactory for Evaluation Slides as Normal Ones

Sampling issues are a common reason for FN cytology results. In studies on the reevaluation of NILM slides preceding cancer diagnosis, a fraction of slides were subsequently rated as unsatisfactory for evaluation, according to the Bethesda system [51,53], with inflammation and obscuring blood hampering the proper diagnosis [54,55]; however, the scale of this phenomenon is not clearly established. In the study by Ejersbo et al., 81% of all negative cytologies preceding CC diagnosis were recognized as sampling errors after a review [46]. According to Sherman et al., among 18 patients with an overall 123 multiple normal slides preceding CIN3+ diagnosis, 7 had at least 2 slides reclassified as unsatisfactory for evaluation (38.9%) [51]. In an audit of FN in Poland, only 3 amongst 48 re-evaluated normal slides preceding CC diagnosis were reclassified by all three experts as unsatisfactory for evaluation (6.3%) [45]. 

##### True Lack of Abnormal Cells on the Slide

Some FN slides truly do not contain abnormal cells. This may be related to the absence of CC precursor lesion on the cervix at the time of sampling and extremely short time of cancer development and invasion [56], as well as the nature of the lesion and its location deep in endocervical canal or under the mucosa, which impede proper sampling [57]. The risk factors of rapid onset of CC are not well-investigated; it was shown by Hildesheim et al. that they are non-specific and similar to those conditioning a typical development of the disease [58].

##### Histology of the Lesion

Adenocarcinoma (ADC), a histological subtype of CC, was identified significantly more frequently than squamous cell carcinoma (SCC) among women with FN cytological results, compared to those with screen-detected cancer, due to their common location in the endocervical canal and more difficult sampling [57,59,60,61]. Based on the audit of over four thousand CC cases diagnosed in Sweden in 2002–2011, Wang et al. concluded that women with normal screening results had an 89% lower risk of SCC and only 60% lower risk of ADC diagnosis [62]. The risk of obtaining a FN result, compared to a true positive cytology report, was shown to be over three times higher in women subsequently diagnosed with ADC, compared to SCC, and almost two times higher in those with diagnosis of carcinoma other than squamous and glandular type [61]. It also coincides with the trend of the raising rate of ADCs detection in countries with CC screening implemented; since this type of carcinoma is more difficult to diagnose in screening, the treatment of precancerous lesions is suspended, and they may develop to the fully invasive form of disease [63].

Overall, the lower sensitivity of cytological screening for the detection of ADC than SCC is a complex issue and may result from a true lack of abnormal glandular cells on the slide due to: (1) the location of the lesions and less efficient sampling [64], (2) difficulties in the proper identification of these cells on the slide and abnormal features [65,66], and (3) the possibly faster development and higher rate of progression of ADC precursors [67]. Data on the performance of cervical cytology in screening for cases of histology, other than SCC and ADC, are scarce [61,68,69]. 

##### Limited Reproducibility

Cytology is an examination with limited reproducibility. As shown by Stoler et al. in the ALTS study (ASC-US/LSIL Triage Study) on almost five thousand cytological slides, the kappa coefficient was only moderate (κ = 0.46, 95% CI 0.44–0.48) for an atypical squamous cell of undetermined significance (ASC-US) and low-grade squamous intraepithelial lesion (LSIL) cytologies [70]. Sørbye et al. found similar agreement on a set of 100 slides, with uniform distribution of NILM, ASC-US, LSIL, atypical squamous cells—cannot exclude high-grade intraepithelial lesions (ASC-H) and high-grade intraepithelial lesions (HSIL) results, with the least agreement in low-grade abnormalities [71]. In the ATHENA study, four laboratories in the United States evaluated over forty-six thousand LBCs; despite the similar background population, the differences between them in the rates of abnormal results were substantial (3.8–9.9%) [72]. All these results confirm the subjectivity of cytological evaluation, which may also impact the rates of FN results.

#### 2.1.2. High Risk HPV Tests 

Depending on the oncogenic potential of specific HPV genotypes, the International Agency for Cancer Research (IARC) classified 12 HPV types (16, 18, 31, 33, 35, 39, 45, 51, 52, 56, 58, and 59), as associated with a high risk (HR) of CC development. Eight types (26, 53, 66, 67, 68, 70, 73, and 82) are considered probably carcinogenic due to the more common occurrence in CC cases than in cohorts with normal cytology [73]. The oncogenic potential of the subsequent nine types (6, 11, 34, 40, 42, 43, 53, 54, and 73) is classified as low [74].

As a primary prevention against HPV infection, prophylactic HPV vaccines are currently available, and many countries worldwide have decided to implement HPV vaccination into national immunization programs [75]. The bivalent vaccine against the HPV-16 and -18 types is being gradually replaced with quadrivalent (against HPV types 6, 11, 16, and 18) and nine-valent (against HPV types 6, 11, 16, 18, 31, 33, 45, 52, and 58) vaccines. All of them were shown to be safe and effective [76,77,78], with the highest efficacy in the prevention of precancerous lesions reaching 100%, when vaccinating girls without previous contact with the virus [79,80].

Only two likely carcinogenic genotypes (66 and 68), according to IARC classification, are included in the standard HPV tests used in the primary screening. Due to the rare occurrence of remaining genotypes among CC cases (26, 53, 67, 70, 73, and 82), a possible slight increase in sensitivity at the expense of a high decrease in specificity, extending the spectrum of HPV types included in validated screening tests, is not recommended [81,82]. 

According to the meta-analysis by Koliopoulos et al., the sensitivity of HR HPV tests for the detection of CIN2+ lesions in group of women aged 30 or more was 93.9%, compared to 72.2% for cervical cytology (both LBC and Pap). HPV tests were shown to be statistically significantly more sensitive for CIN2+ detection than both conventional cytology and LBC, with relative sensitivities of 1.52 and 1.18, respectively. Specificity was significantly lower for HPV testing (with relative specificities of 0.94 for Pap test and 0.96 for LBC). Similar results were obtained for CIN3+ detection [24]. 

On account of the increasing number of HPV tests on the market without proper validation, some criteria ensuring safety and high-quality performance were urgently needed [83]. In 2009, Meijer et al. published a list of requirements to be fulfilled by the producer, in order to introduce the test for primary HPV screening, which included: (1) a sensitivity of at least 90% of sensitivity of the Hybrid Capture 2 test for CIN2+ detection in women ≥30 years old; (2) a specificity of at least 98% of specificity of the Hybrid Capture 2 test for CIN2+ detection in women ≥30 years; and (3) the intra-laboratory reproducibility and inter-laboratory agreement, with a lower confidence bound of at least 87% [84]. According to recent study by Poljak et al., 254 distinct HPV tests and over 425 test variants exist on the market; however, over 90% of them were not validated in accordance with Meijer’s criteria [85]. 

The use of HPV tests for detecting HR HPV types in screening significantly reduced the problem of FN results, compared to cytological-based screening, due to the significant increase in sensitivity [24]. Despite the high sensitivity, the number of women at risk of FN results in HPV testing increased as the method spread around the world, and this should be carefully monitored [18]. The reasons for a truly negative HPV test result preceding CC diagnosis include the following: (1) the true lack of HPV DNA related to the non-HPV-associated nature of cancer; and (2) the histological misclassification of endometrial cancer as of cervical origin. FN results may potentially follow from: (1) cancer development from infection with low-risk oncogenic types; (2) clearance of HPV infection before sampling leading to the conclusion of HPV-unrelated disease; and (3) sampling and laboratory errors.

##### A. Reasons of Negative HPV Test Results in CC Cases

True lack of HPV DNA in the sample

The homogenous group of SCC, which accounts for about 75–85% of all CC cases worldwide [86,87], is highly associated with HPV infection [88]. The second type, ADC, is rarer, with many subtypes distinguished and some of them being extremely rare [89]. The overall rate of HPV-negative (HPV(−)) ADCs ranges between 15% and 48% and strongly depends on the subtype [85,90,91,92,93]. A low viral load in all ADC types additionally impedes the capability of its detection; as a consequence, the diagnosis is stated later, and the detected tumors are larger [94].

According to the International Endocervical Adenocarcinoma Criteria and Classification, two groups of ADCs were extracted, depending on the HPV infection status: (1) non-HPV-associated (endometrioid, gastric type, serous, clear cell, and mesonephric carcinoma) and (2) HPV-associated (usual, villoglandular, mucinous (not otherwise specified), mucinous intestinal, mucinous signet ring, and invasive stratified mucin-producing carcinoma) [95]. Detailed information on HPV-dependent and -independent CC types is presented in Table 2, based on the current WHO Classification of Tumors [5]. 

In general, HPV(−) CC cases were shown to be associated with higher age, worse prognosis, diagnosis of ADC, and higher risk of relapse and distant metastases [85,96,97,98,99].

Histological Misclassifications

Histological misclassifications seem to explain a part of the HPV(−) CCs phenomenon. Endometrial cancer, which is not, or barely, related to HPV infection [94], may be misclassified as being of cervical origin. This erroneous classification may also partially explain the higher age of HPV(−) CC cases, compared to HPV-positive (HPV(+)) ones, since the risk of endometrial cancer is the highest in women around 70 years old [100,101], which is about 10 years later than the peak of CC incidence.

In the study by the Cancer Genome Atlas Research Network, 8 out of 178 primary CC cases were reclassified as endometrial-like (5%); a total of 7 of them were HPV(−) [102]. In a study of 371 biopsy-proven CCs, 21 of 31 HPV(−) cases were pointed out as being of non-cervical origin (68%) [103]. Lower rates of misclassified CCs were reported by Pirog et al. in the retest of 760 ADC cases collected worldwide [93]: 49 of 731 cases (6.7%) were reclassified by the board of four pathologists: 23 of them (3.1%) were classified as not, or doubtfully, being of cervical origin, with 26 (3.6%) classified as non-epithelial neoplasias. Indeed, inter- and intra-rater agreement in the histological classification of non-SCC depends on the type of carcinoma and is rather low, with a kappa coefficient (measuring the agreement beyond the chance) of 0.44, indicating moderate agreement, according to Landis and Koch scale [104,105]. However, another study showed low reproducibility in HPV-associated ADCs and higher reproducibility for non-HPV-associated ADCs [106].

##### B. Reasons of Potential FN HPV Test Results in CC Cases

Cervical Cancer Developed from Infection with Low-Risk Oncogenic Types

Published studies confirmed the relationship between the HPV types classified by WHO as probably, or possibly, carcinogenic (which are not being detected by primary HPV screening tests) and the development of a fraction of CC cases. Those cases are usually single-infected, which makes them impossible to diagnose with contemporary HPV screening tests. 

In the ATHENA trial, with over 46 thousands women involved, 55 HPV(−) CIN2+ cases were found [107]. After re-evaluation of the LBC samples with more sensitive tests, 22 cases were reclassified as high-risk HPV(+) (40.0%), and the next 10 were reclassified as low-risk HPV(+) (18.2%) with HPV-73 or HPV-82 detected. Halec et al. showed the transcriptional activity of HPV types 26, 53, 66, 67, 68, 70, 73, and 82 in cases with CC and single HPV infection of these types [108]. The HPV-73 type was also confirmed as oncogenic by Amaro-Filho et al. [109]. Of 544 CIN3+ cases with a single infection in Japan [96], the HPV types listed by Halec et al. were found in 28 cases, with single infection being only 5.1%. Another study of 136 CC cases showed the reclassification of 6 out of 14 (42.9%) primary HPV(−) women as HPV(+), with HPV-11, 16, 18, 45, and 68 detected [97]. In a study of the Latin USA population, single HPV-90 prevalence was unexpectedly high, with 9.4% of infected women having additional cytological abnormalities. Most women who were diagnosed with HPV-90 (96.8%) had a single infection [110]. In the study of 1739 patients with SCC, 14 women (0.8%) were infected with one of the WHO-probable carcinogenic HPV types, and all of them were reported as single-infected [111].

Hit and Run Theory

A hit and run theory may bring another explanation for HPV(−) CCs phenomenon. The hypothesis is broad and related not only to CC. It states that some types of cancer need the viral infection to initiate the oncogenesis; however, as the mutations are accumulated, the need for oncogenic factor presence no longer exists, and the initial infection clears [112]. Subsequent cancer is, indeed, viral-driven, but the test performed after clearance is unable to detect absent viral DNA. 

According to the work by Tjalma et al. on over 6000 CIN2+ women, comparing HPV prevalence in high-grade CIN and invasive CC, the rate of HPV(−) high-grade CIN cases was lower than the rate of HPV(−) invasive CCs (2% vs. 8%, respectively) [91]. Differences were even sharper in a study by Coutleé et al.: 11.5% of invasive CCs were HPV(−), whilst only 0.2% of CIN2 and CIN3 women no detected HPV DNA [113]. This observation stayed in line with the hit and run hypothesis, as the longer time needed for invasion may also lead to HPV infection clearance. However, this hypothesis clearly needs further investigation. Studies should be planned, in order to prove the absence of HPV DNA in invasive cancer cases, as well as the presence of HPV DNA and expression of genes involved in carcinogenesis in the precursor lesions sampled prior to cancer diagnosis.

Sampling Errors, Laboratory Errors

Aside from the test-specific factors, other common mistakes may be responsible for FN results. Sampling errors may result in a lack of the necessary amount of HPV DNA material; a similar problem is vital in cytological screening [42]. As self-sampling methods for HPV detection emerged, the question was raised regarding whether the rate of unsatisfactory for evaluation self-collected samples will be acceptable. Recently published studies suggested that proper test selection for self-sampling results in a sensitivity comparable with clinician-sampling; therefore, sampling errors will be of less importance [114]. 

Another classical reason for obtaining false-negative results is laboratory error, following inappropriate laboratory procedures, poorly trained staff, no validation or quality assurance of procedures, and the use of screening tests with no analytical and clinical validation [84,115].

#### 2.1.3. Visual Inspection with Acetic Acid

In settings where financial resources and the number of well-trained health workers are limited, the WHO recommends using VIA with the immediate treatment of suspected treatable lesions as a screening method. VIA involves the application of a 3% to 5% solution of acetic acid on the cervix and observation of the acetowhite lesions using the naked eye after at least 1 min since application. If necessary and appropriate, treatment with cryotherapy or loop electrosurgical excision procedure (LEEP) should be provided. This approach is aimed at ensuring a high negative predictive value, reducing the loss to follow-up, and treating all women in need using scant finances [25].

The sensitivity of VIA screening for CIN2+ detection ranged widely between 42% and 92% in 26 studies, where all subjects underwent confirmatory testing, with a pooled sensitivity of 80% [116]. Similar results were shown in a meta-analysis by Arbyn et al. [117]. The results, however, may be overestimated, due to some methodological issues [118]. Firstly, a fraction of studies on VIA effectiveness for symptomatic women were included [116]. Secondly, colposcopically-targeted biopsy was most often used as a gold standard, and this method is correlated with VIA results, since both of them rely on a visual evaluation [117,119]. Thirdly, in many settings, women were screened for the first time in their lives, which may result in the easier detection of more advanced lesions [120,121].

The phenomenon of FN VIA results is not widely discussed in the literature, due to the lack of long-term follow-up of women in developing countries. Besides, the gold standard of colposcopy with biopsy is not always performed in all screened women; in many studies, only VIA(+) participants underwent confirmatory testing [116]. The potential causes of false-negative VIA results may include: small size, endocervical and submucosal location of the lesions, and various reactions of the abnormal epithelium on acetic acid application; however, these have not been well-investigated in the literature on the subject. Low reproducibility and observer-dependence are the next reasons.

VIA, as an unquantifiable method based on the human eye evaluation only, is strongly observer-dependent and has low reproducibility. The results of the study in rural Nigeria showed that, depending on the provider, 0–21% of women were suspected to have cancer, and 0–25% of women were assessed as VIA(+), despite the homogenous background population [120]. In the study carried out in Andhra Pradesh, India, women were examined by six gynecologists, and their positivity rate varied between 4% and 31% [121]. Training plays an important role in ensuring the optimal performance of VIA-based screening [122]. In a study on over 140 thousand of women in India, retraining resulted in a drop in positivity rates from 17% to 10%. Additional training entailed an increase in the agreement of both cytological and histological evaluations, as well [123]. 

Additionally, the definition of the disease threshold may impact the accuracy of VIA, as it is not strict. As stated in guidelines by JHPIEGO, abnormalities looking similar to “raised and thickened white plaques or acetowhite epithelium, usually near the SCJ (squamocolumnar junction)” are of clinical significance when using VIA for screening, as well as “cauliflower-like growth or ulcer; fungating mass”, which should be considered a cancer suspicion [124]. Cryotherapy is recommended if the whole SCJ is visible, the lesion is fully visible, and it does not exceed 75% of the ectocervix [25]. All of these criteria are ambiguous, and the impact of human subjective evaluation cannot be completely excluded, regardless of the degree of detail in the description. The need for the appropriate and regular training of health workers providing VIA is, therefore, essential. Additionally, the use of artificial intelligence is currently discussed as a helpful tool for distinguishing clinically significant lesions from others [125].

### 2.2. Triage Methods

#### 2.2.1. Colposcopy with or without Colposcopically-Directed Biopsy

Colposcopy is one of the most commonly used triage methods to screen positive women in cytology and HR HPV-based screening [126]. Colposcopy protocols included the comprehensive evaluation of the cervix, with the use of the magnified appearance after the application of an acetic acid solution and Lugol’s iodine, identification and initial grading of abnormalities, collection of tissue material for histological examination, and excisional procedures, if necessary [127,128]. 

Despite the numerous attempts to standardize the procedure for the proper training and certification of colposcopists, the accuracy of the procedure is far from perfect, and FN results are common. According to the meta-analysis by Mitchell et al., the pooled rate of positive cone biopsies, widely accepted as a gold standard, among colposcopically-directed negative biopsies in women with cytological abnormalities reached approximately 4% [129]. In a prospective study of follow-up women with cytological abnormalities and negative colposcopy, 9% were subsequently diagnosed with CIN2+ [130]. A follow-up of participants of ALTS study with adequate baseline colposcopy revealed 30% of them were diagnosed with CIN2+ over the 2 years since their baseline test [131]. 

The rate of CIN2/CIN3 detected by colposcopy with targeted biopsy reached 56% in 2112 women with cytological abnormalities [132]. On the other hand, according to meta-analysis by Underwood et al., over 91% of CIN2+ women were correctly identified, but the authors admitted that high sensitivity values probably followed from verification bias. When restricted to the studies with immediate excisional treatment following colposcopy, sensitivity dropped down to 81.4% for CIN2+ detection [133]. Underestimation of diagnosis in colposcopically-directed biopsy is especially common in women subsequently diagnosed with microinvasive CC [134].

##### Low Reproducibility

Likewise in cytology, colposcopic examination, as a method based on the evaluation by human eye, is subjective and observer-dependent. Poor agreement between colposcopists was reported in the literature [132,135] with lowest agreement for evaluating low-grade lesions [70,135,136]. Low reproducibility (with kappa coefficient of about 0.2) for critical issues impacting the final result—sharpness of margins, atypical vessels, or lesion presence—was also demonstrated [137,138]. In a study by Pretorius et al. 7 physicians achieved sensitivity of colposcopy with colposcopically-targeted biopsy between 28.6% and 92.9% for CIN3+ detection [139]. The size of the lesion and its severity was shown to have an impact on accuracy of the colposcopic impression—it appeared more precise when more quadrants of cervix were affected by the lesion [140].

Sensitivity of colposcopy is believed to depend on colposcopists’ experience; however, many studies contradict this statement [131,132,141]. Gage et al. reported similar sensitivity for nurse practitioners, general gynecologists, gynecologic oncology fellows and gynecologic oncologists [131]. Bekkers et al. demonstrated higher sensitivity for junior compared to senior colposcopy fellows but at the expense of lower positive predictive value which is a logical effect of less self-confidence resulting in tendency of upgrading diagnoses and taking more biopsies by less experienced colposcopists [142]. Colposcopy performance improvement may be achieved, however, by providing standardized training to colposcopy fellows [143,144]. For the last decade, efforts have been made for developing standardized colposcopic terminology by, i.e., the International Federation for Cervical Pathology and Colposcopy (IFCPC) [145] and American Society for Colposcopy and Cervical Pathology (ASCCP) [128,146]. Still, further work is needed to disseminate the proposed nomenclature and train medical staff. 

##### Discrepancies between Actual Lesion Severity and Colposcopic Image

The issue of colposcopically-targeted and random biopsies taken during colposcopy was widely discussed in the literature. Since the severity of lesion did not always correspond with the most visually advanced area of the colposcopic appearance, random biopsies were suggested, in order to elevate colposcopy sensitivity [147,148]. The study by Pretorius et al. reported that 37% of CIN2+ cases were recognized by random biopsy and not by colposcopically-directed biopsy [149]. Huh et al. analyzed the results of 2796 women who participated in ATHENA study, with random biopsies performed during colposcopy, due to no visible lesions on the cervix: 20.9% and 18.9% of biopsies were subsequently diagnosed as CIN2+ and CIN3+, respectively [147]. In the study by Gage et al., taking two or more biopsies during one procedure was found to significantly improve the sensitivity of colposcopy [132], which was confirmed by others [139]. Endocervical curettage was shown to improve sensitivity of colposcopy performance, with the yield of 2% to 15% for CIN2+ detection [149,150,151].

##### Histological Discrepancies/Misclassifications

A part of the FN colposcopies phenomenon may also occur due to histological misclassifications. According to a study by Ismail et al. based on 100 colposcopic biopsies, as evaluated by 8 experienced colposcopists, the overall agreement was moderate but highly dependent on lesion severity, ranging from a kappa of 0.175 for CIN1 and CIN2 to 0.832 for invasive cancer [152]. In a similar study by McCluggage et al., the average of overall agreement between pathologists was poor, with a kappa of around 0.3 [153]. In a histological review in Poland, only 205 out of 368 histological samples that were primarily assessed as CIN2+ were subsequently confirmed in an expert diagnosis (55.7%) [154]. However, the discrepancies resulting in normal- and high-grade result were rather rare and, based on the mentioned studies, occurred in <1% of paired evaluations.

#### 2.2.2. P16/Ki-67 Dual Staining

DS for the simultaneous detection of p16 and Ki-67 proteins within the same cell became a widely investigated triage procedure, due to its high sensitivity, relatively high specificity, and possibility to perform on the residual sample after the HPV test and LBC [155]. In the DS procedure, at first, the cellular material is stained to visualize cells containing both proteins. Next, the slide is prepared for microscopic evaluation. The threshold for positivity is predominantly set at 1 dually-stained cell.

DS was shown to have a higher sensitivity in detecting precancer lesions and provide longer negative reassurance in HPV(+) women than negative cytology, with 5 years relative risk of CIN2+ detection of 0.7 [156,157]. In published studies, the sensitivity for CIN3+ detection was stable and achieved about 92% [155,157,158]. The sensitivity for CIN2+ detection seemed to not depend on the examined population, either. For women with ASC-US/LSIL, the sensitivity was 88% [158]; for HPV(+) women, it was 86% [155,159], and for the HPV(+) cohort with abnormal cytology, it was 85% [156]. In a meta-analysis by Peeters et al., 84% sensitivity and 77% specificity were reported for CIN2+ detection, with similar values for CIN3+ detection in the ASC-US+ women (88% and 72% for sensitivity and specificity, respectively) [160]. 

##### Slide Misinterpretation

Published evidence pointed out difficulties in slide interpretation as a reason of improper DS result. As the threshold for positivity is usually set at one dual-stained cell on the slide, the correct exclusion of the positivity of each visible cell is crucial. Cells presented in groups on the slide, or with a scant cytoplasm and weak p16 staining, seemed to be challenging for readers [161,162], as well as, on the contrary, those with extremely strong background staining, which hampered appropriate evaluation of nucleus staining [163]. A low number of dually stained cells, sometimes with only one visible, may occur and lead to oversight [163,164]. The level of subjectivity in evaluation may be decreased, and the total agreement can be increased by training [163,164,165]. Nevertheless, the reported reproducibility is good to excellent for interobserver agreement, with kappa values ranging between 0.65 and 0.91 [162,163,165].

The issue of factors influencing overall positivity of a DS slide is by far better investigated in the literature than the matter of strict FN DS reports. It was shown by Benevolo et al. that the longer time between sampling and immunostaining or between immunostaining and evaluation of the slide, the higher probability of its positive interpretation. Other factors like referral cytology result, presence of CIN2+ or HPV mRNA detection were also naturally shown as associated with slide positivity [166]. To our best knowledge no other reasons of possible FN reports for DS were discussed in published reports.

#### 2.2.3. Methylation of FAM19A4 and hsa-miR124-2 Genes

The test measuring methylation levels of specific genes involved in the process of carcinogenesis was shown as a promising method of precancerous lesions detection in HPV(+) women. Many potential biomarkers have been verified to find appropriate ones [167,168,169,170]. According to the data published in 2020, four methylation tests of different genes are currently commercially available on the European market for cervical cancer detection: two of them to be performed on cervical smear: GynTect^®^ (based on genes: ASTN1, DLX1, ITGA4, RXFP3, SOX17, and ZNF671) and QIAsure Methylation test kit (FAM19A4 and miR124-2 genes) and two on cervical scrapes: PAX1 DNA detection kit (PAX1 gene) and ZNF582 DNA detection kit (ZNF582 gene) [171]. The performance of the assay detecting the hypermethylation of FAM19A4 and hsa-miR124-2 was demonstrated as promising: at first in research on cell cultures [172,173] and afterwards in humans [174,175] and at present this test seems to be the best validated one.

In published studies virtually all CC cases were properly identified by methylation tests based on these genes. In a cross-sectional study of 519 CC cases from 25 countries in 5 continents, 510 (98.3%) had positive methylation test result and—worth noting—the set of samples consisted of both cervical scrapes and cervical tissues and covered SCCs and ADCs as well as adenosquamous cell carcinomas and other rare CC histotypes [176]. Additionally, HPV(−) cases were correctly detected by the methylation test (18/19, 94.7%). The results of smaller studies performed on both cervical scrapes and clinician-taken samples [174,175,177,178], as well as on self-samples [179,180], stayed in line with the results of Vink et al., pooling all results of 208 out of 213 CC cases, which were detected by the methylation test (97.7%), including 31/31 for self-samples (100%) and 177/182 for other samples (97.3%). The sensitivity for detecting CIN2+ and CIN3+ lesions, however, was lower and ranged between 56.1% to 80.3% for CIN2+ lesions and between 69.6% to 94.7% for CIN3+ lesions [175,178,180].

##### Lower Risk of Progression

The negative result of methylation test in women with CIN2 or CIN3 may indicate low risk of progression to cancer in these persons. De Strooper et al. used the length of preceding HPV infection in women with precancerous lesions as a proxy for advancement of the lesion and risk of progression to cancer. In his study all CIN2+ lesions preceded by HPV infection lasting >5 years were detected by the methylation test (51/51) and only 8/19 (42%) of early CIN2/CIN3 lesions (with <5 years since prior HPV infection) were detected [174]. Similar results were published by Bierkens et al. for CADM1 and MAL genes; levels of methylation of those genes in women with CIN2/CIN3 were higher for cases with ≥5 years of preceding HPV infection compared to those with <5 years of preceding HPV infection [181].

This explanation of lower performance of methylation test in less advanced lesion is in line with observation that CIN2/CIN3 lesions were more commonly methylation negative in women below 30 years old [182,183]. Indeed, the regression rate in younger women is higher and is probably caused by a shorter time of associated HPV-infection [184]. However, the lower sensitivity in detecting CIN2/CIN3 lesions in young women may be clinically beneficial by applying surveillance instead of treatment in women with lower risk of cancer progression. 

##### Borderline Methylation Level

Samples with methylation level near the cutoff value seem to be less reproducible. Bonde et al. observed that 80 samples among 983 selected for both intra- and inter-laboratory agreement analysis had discordant results (8.1%), with 77 of them being close to the methylation positivity threshold (96% of all samples with discrepant results). Additionally, the only case of CC undetected by methylation test in this study had borderline methylation level [178]. This phenomenon may be responsible for both FN and FP results and indicate the need of establishing cutoff properly.

## 3. Prospects in Cervical Cancer Screening

In order to address still unmet needs in CC screening some novel methods have been proposed. HPV testing on first-void urine samples seems to be an acceptable method for reaching underscreened women, with sensitivity and specificity comparable with respective indicators of HPV test performed on clinician-taken samples [185,186]. HR HPV digital droplet polymerase chain reaction test, based on the fractionation of a sample into droplets and subsequent detection and quantification of HPV DNA in each of them, may improve the sensitivity of HPV detection in CIN lesions and early CC tumors [187,188]. However promising, yet more research is needed for safe implementation these methods into real-life CC screening. 

### 3.1. Implications for Screening

To balance sensitivity and specificity of screening process both indicators of the test as well as interval between screens should be taken into consideration. More frequent testing increases sensitivity and reduces FN rate at the expense of decreasing specificity and rising number of FPs. Risks of both CC and precancer occurrence after a negative screening test result were investigated in studies with large sample size and long-term follow-up to assess negative predictive value of the test and establish acceptable intervals [189]. Nevertheless, in some countries, the opportunistic screening without any restrictions on the interval competes with the organized program. Opportunistic screening is, however, not recommended, as it leads to a high smear consumption in only a part of population, limits the accessibility to screening for some women, and may result in poor quality of offered procedures and low cost-effectiveness [20]. Only organized screening programs with intervals set according to the risk of FN and FP results enable limiting these adverse screening outcomes. 

#### 3.1.1. Screening Tests

No screening or triage test ideally distinguishing healthy and diseased people exists, neither for CC screening nor for any other type of screening. Sensitivity of 100% is beyond the reach of current possibilities and both policymakers and screening organizers should be aware of it. Women must be informed that participation in screening significantly reduces the risk of both incidence and mortality of CC but it does not eliminate it completely. Emphasis should be placed on the use of certified and validated screening tests, only to ensure highest possible quality. Additionally, quality assurance procedures and quality control should be in place.

It is impossible to avoid FN results in cervical cancer screening but due to their serious consequence efforts should be undertaken to limit their burden. The whole set of reasons of FN results may be divided into broadly understood errors and test deficiencies. To minimize human factor extensive and regular training should be introduced, however, it does not always guarantee satisfactory effects and may be difficult to run in some settings. According to the “WHO guidelines for screening and treatment of cervical pre-cancer lesions for cervical cancer prevention”, HPV testing is considered superior to both cytology-based and VIA screen-and-treat approach by means of overall effectiveness of screening program [25]. HPV-based CC screening is less subjective than cytology or VIA, may be automated, run on a large scale of self-collected material, or as a part of point-of-care infrastructure and “see-and-treat” protocols. Its main advantage is the minimization of FN result rate. While comparing the cytology currently being replaced with the HPV test, in the sense of FN results rate for CIN2+ detection, the latter seem better, with about 7% of potential false HPV(−) results, compared to 13–70% for the Pap test [36]; a similar comparison can be made for VIA as well, with the rates ranging between 8% and 58% [72].

Although HPV-based screening is very sensitive, false-negative results may occasionally occur [190]. As supporters of co-testing state, adding cytology to the primary HPV testing increases the screening sensitivity and gives the opportunity to detect HPV(−) subtypes of ADC, which are invisible for HPV testing. However, ADCs cause considerable difficulties for cytological detection and even co-testing may be insufficient for detecting them [60]. The study on over 1.2 million co-tested women showed that 3.5% of precancers and 5.9% of cancers were initially HPV(−) and cytology-positive [190]. However, bearing in mind the low rate of CC incidence, the yield of using co-test, compared to the HPV test alone is approximately only five additional cases detected per million screened. In the light of hit and run hypothesis at least some of precancer lesions should be successfully detected by the HPV test before infection clears and HPV(−) CC develops. Keeping in mind also high costs of cytology performance [191] and maintenance of cytological infrastructure, the use of primary HPV testing alone seem more reasonable than co-testing.

#### 3.1.2. Triage Methods

A relatively low specificity of HPV-based screening, however, leads to an excessive number of referrals for colposcopy, colposcopies performed, and treatment procedures executed. Accurate triage methods are, therefore, crucial for avoiding both overdiagnosis and overtreatment. Some actions were taken to support proper decision when using subjective screening and triage tests; this may both limit the number of FN and FP results. 

Colposcopy still serves as a gold standard in many countries with screening programs implemented worldwide. A few decades ago, cervicography, based on taking pictures of the cervix, which were subsequently evaluated by trained colposcopists, was proposed as a useful tool [192]. More recently, attempts were made to digitize cervigrams, in order to facilitate their storage and usage [193]. The combination of deep-learning algorithms on collected cervical images and the medical history of patients allowed us to develop automated visual evaluation (AVE) application that supports providers in proper decision-making, regarding patients’ management, especially in LMIC. The results seemed promising [194]. However, further research is needed for its proper validation before its implementation into real-life screening [125]. Additionally, appropriate nomenclature and colposcopic protocols were developed as a response to the need for the unification of standards [127,128]. Still, the further need for their dissemination and implementation in screening programs is vital.

DS obtained approval by the U.S. Food and Drug Administration with the indication for HPV(+) women with HPV genotypes other than 16 and 18, in order to determine the need for colposcopy referral, as well as for HPV(+) women with HPV 16 or 18 detected, to guide their management [195]. There have been some endeavors to automize and objectify DS assessment. Wentzensen et al. showed that DS using artificial intelligence (AI) and deep-learning protocols had a lower positivity rate, equal sensitivity, and higher specificity, compared to both manually evaluated cytology and DS [196]. However, these methods need further validation.

Among the investigated triage procedures, the methylation of FAM19A4 and miR124-2 genes provided promising results, in terms of high sensitivity for cancer detection, and allowed for the use of self-sampled vaginocervical material. However, the rate of FN methylation results is higher for less severe CIN2 and CIN3, which are the main targets of CC screening. According to the aforementioned studies, the methylation test correctly identifies cases with long-lasting HPV infection; however, further studies are required, in order to assess the potential for the progression, regression, and persistence of HPV(+) methylation-negative HSIL cases. 

## 4. Conclusions

To conclude, FN results are unavoidable in screening, and this phenomenon can only be minimized by careful quality assurance. Human error can be reduced, but not excluded, by both training and use of AI methods. In light of the published studies and despite some inevitable adverse effects, HPV-based screening seem to be the most efficient, in regard to FN rates. The triage methods of HPV(+) women are still being investigated. Both DS and methylation of FAM19A4 and miR124-2 genes seem promising for the future; however, colposcopy with appropriate protocols and quality measures incorporated will probably still serve as a reliable triage examination, until further studies will be performed to elaborate on the most optimal protocol for proper risk stratification and management of HPV(+) results.

## Figures and Tables

**Table 1 diagnostics-12-01508-t001:** The basic parameters of screening tests and triage methods, with potential reasons of false negative results occurrence. Sensitivity and specificity data were retrieved from comprehensive meta-analyses. Pooled rates were reported with range of parameter analyzed in brackets. For dual staining, 95% CI from meta-analyses were given in brackets. Cytology parameters were reported for atypical squamous cell of undetermined significance benchmark; * for punch biopsies taken in CIN1+ women; ** considering studies with excisional treatment following punch biopsies only; *** for women with low-grade cytological abnormalities (atypical squamous cells of undetermined significance, low-grade squamous intraepithelial lesions).

Examination	Sensitivity (%)		Specificity (%)		Potential Reasons for False Negative Results Occurrence
	CIN2+	CIN3+	CIN2+	CIN3+	
**Pap test**	66 (34–96)	70 (39–85)	96 (86–99)	97 (85–99)	(1) misclassification; (2) sampling errors; (3) true lack of abnormal cells; (4) subjectivity of evaluation.
**Liquid-based cytology**	76 (52–94)	76 (52–98)	92 (77–97)	92 (73–97)	
**High-risk HPV test**	93 (61–100)	97 (81–100)	89 (64–95)	89 (69–94)	(1) true lack of HPV DNA in sample, due to non-HPV associated histological type of lesion; (2) histological misclassification of endometrial cancer as cervical cancer; (3) cancer developed from HPV type not detectable by screening tests (probably carcinogenic, low-risk types); (4) clearance of HPV infection before sampling; (5) sampling and laboratory errors.
**Visual inspection with acetic acid**	79 (65–91)	83 (58–95)	85 (74–95)	84 (74–94)	(1) low reproducibility, observer-dependence; (2) inaccurate definition of disease threshold.
**Colposcopy with or without colposcopically-targeted biopsy**	91 (56–100) * 81 **	91 (75–100) *	25 (0–79) * 63 **	18 (0–65) *	(1) low reproducibility, observer-dependence; (2) discrepancies between lesion severity in terms of histology and colposcopic image; (3) histological misclassification of biopsies taken during colposcopy.
**p16/Ki-67 dual staining *****	84 (77–89)	88 (58–98)	77 (70–82)	72 (67–76)	(1) slide misinterpretation.
**Methylation of FAM19A4 and miR124-2 genes**	68	79	78	77	(1) truly low risk of profession of precancerous lesion to cancer resulting in a negative methylation result interpreted as FN; (2) true lack of abnormal cells in a sample; (3) borderline methylation level.

**Table 2 diagnostics-12-01508-t002:** HPV-dependency in cervical cancer types, according to the WHO Classification of Tumors; NOS—not otherwise specified.

HPV-Dependance	Precursor Lesions	Invasive Lesions
**HPV-associated**	Squamous intraepithelial lesions: LSIL (condyloma/CIN1) HSIL (CIN2/CIN3)	**Squamous cell carcinoma:** non-keratinizing keratinizing basaloid warty (condylomatous) papillary lymphoepithelioma-like
	Adenocarcinoma in situ	adenocarcinoma, usual type villoglandular adenocarcinoma
		**Adenocarcinoma, mucinous type:** mucinous NOS adenocarcinoma intestinal adenocarcinoma signet-ring cell adenocarcinoma
	Stratified mucin-producing intraepithelial lesions	**Adenocarcinoma, mucinous type:** stratified mucin-producing carcinoma
	Unknown	carcinosarcoma
		adenosquamous and mucoepidermoid
		adenoid-basal carcinoma
		**Neuroendocrine neoplasia:** neuroendocrine tumors, NOS small cell neuroendocrine carcinoma large cell neuroendocrine carcinoma carcinoma admixed with neuroendocrine carcinoma
**Non-HPV-** **associated**	Mesonephric remnants	adenocarcinoma, mesonephric type
	Atypical lobular glandular dysplasia	adenocarcinoma, gastric type
	Endometriosis	**Adenocarcinoma, NOS:** endometrioid serous
	Unknown	squamous cell carcinoma, HPV-independent
		adenocarcinoma, clear cell type
		adenosarcoma
		**Genetic tumor syndromes:** Perez–Jeghers syndrome Carney complex
**unknown**	Unknown	haematopoetic proliferations and neoplasia
		**Mesenchymal tumors:** solitary fibrous tumor, NOS NTRK (neurotrophic tyrosine receptor kinase)-rearranged spindle cell neoplasm alveolar soft part sarcoma Ewing sarcoma
		**Melanocytic lesions:** mucosal melanoma

## Data Availability

No new data was created or analyzed in this study. Data sharing is not applicable to this article.

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
