# Peer review of "False Negative Results in Cervical Cancer Screening—Risks, Reasons and Implications for Clinical Practice and Public Health"

_diagnostics, 2022, doi:10.3390/diagnostics12061508_

Round 1
Reviewer 1 Report
LINE 573: "AL" FOR "ALL".
Author Response
In line 573 it is not “AL” but “AI” (artificial intelligence). This abbreviation should have been expanded, however. Proper amendment has been made.
Reviewer 2 Report
It is thought to be meaningful as basic data that compared and analyzed the accuracy of cervical cancer screening methods.
However, in many countries, medical systems have already been created to reduce false-negative errors in cervical cancer screening tests.
Since the screening procedure is different for each country, it is thought that the consideration on how to reduce the false-negative and false-positive rates according to the test subject should be supplemented.
For example, methods to reduce screening errors may be different for women who can receive free cervical cancer screening every year and for women who can get cervical cancer screening for more than 5 years and at irregular intervals. I think it is necessary to provide useful information about this situation.
However, as the sensitivity and specificity analysis and possible errors of the cervical cancer screening method are well organized, it is expected to be helpful for clinical application.
Author Response
We would like to thank the Reviewer for this valuable comments. Appropriate paragraph commenting on this topic has been included in the beginning of “Implications for screening” section.
Indeed, the risk of obtaining false negative report in screening differs when different intervals are implemented in the programme. However, the type of screening test should compensate this differences, i.e. when more sensitive method is used, the interval may be extended which is not the case if performing less sensitive test. Clearly, this is the ideal situation when women adhere to the recommended interval which often does not occur, i.e. in opportunistic screening. Opportunistic screening is not recommended by the European guidelines for quality assurance in cervical cancer screening as it may lead to a number adverse effects. Only organised screening programmes with intervals set according to the risk of false negative and false positive result enables limiting these adverse screening outcomes.
Reviewer 3 Report
In the manuscript (diagnostics-1734570) entitled “False negative results in cervical cancer screening – risks, reasons and implications for clinical practice and public health” by Dr. Macios and Dr Nowakowski, a comprehensive literature review on risks and reasons of obtaining False negative results of primary cervical cancer screening tests and triage methods and discuss their clinical and public health impact and implications.
The manuscript is well written; clear, precise, and easy to understand, addressing the topic of clinical misinterpretation of the currently employed cervical cancer detection methods, which is an important topic.
The ms interesting, while the analysis well performed. More supporting literature on the topic should be quoted. I have made several suggestions. Several typo errors should be checked and corrected.
However, it should be noted that the manuscript will improve our general understanding on the negative implication of false negative results during cervical cancer diagnosis, with serious health consequences. In my opinion, the manuscript can be accepted with a minor revision. Please find several suggestions for improving the ms.
Comments
1. The font style sohuld be checked as it changes throughout the text. Please uniform the style
2. Unnecessary highlinings should be removed throughout the text, e.g., line 3, 32, 207, 318, . Please revise the entire text accordingly.
3. I suggest using Italic font for Latinisms, only
4. Line 98 this sub-head title should be in bold style
5. More literature should be included. Several sentences/paragraphs are lacking in supporting references, e.g., introduction, lines 199-203, lines 243-250, as well as other sentences
6. Besides the traditional/clinical cervical cancer screening approaches, novel methods can be, at least briefly, quoted in the manuscript. For instance, the novel and highly reliable High-risk HPV-based droplet digital PCR is acquiring frowing importance in identifying HPV-driven tumors as well as HPV-driven pre-tumoral lesions, including HPV-positive CIN lesions (doi: 10.3389/fmicb.2020.591452 and doi: 10.1007/s13402-017-0331-y). In the future, the ddPCR, might complement current follow-up analyses as well as traditional diagnostic approaches for HPV-driven tumor early identification. This is an important point that, alongside supporting references, should be included.
7. Lines 138-139 the sentence can be improved in terms of readability
8. In the section 2, a brief mention of the currently employed vaccines would be helpful for the reader.
9. Line 224 “remaining genotypes” the unnecessary space between words should be removed. The same consideration can be made in lines 489, 551, 556
10. Line 256 “ADC”, should be Adenocarcinoma (ADC), the first time being quoted. The same consideration can be made for Visual Inspection with Acetic Acid (VIA) in line 360
11. Section 3, line 471 additional genes have been described as differentially methylated in cervical cancer (https://doi.org/10.3389/fgene.2020.00347)
12. Conclusions should be a stand-alone section
Author Response
We would like to thank the Reviewer for important comments. Referring to suggested topics will increase the scientific value of the manuscript.
Response to Reviewer 3:
In the manuscript (diagnostics-1734570) entitled “False negative results in cervical cancer screening – risks, reasons and implications for clinical practice and public health” by Dr. Macios and Dr Nowakowski, a comprehensive literature review on risks and reasons of obtaining False negative results of primary cervical cancer screening tests and triage methods and discuss their clinical and public health impact and implications.
The manuscript is well written; clear, precise, and easy to understand, addressing the topic of clinical misinterpretation of the currently employed cervical cancer detection methods, which is an important topic.
The ms interesting, while the analysis well performed. More supporting literature on the topic should be quoted. I have made several suggestions. Several typo errors should be checked and corrected.
However, it should be noted that the manuscript will improve our general understanding on the negative implication of false negative results during cervical cancer diagnosis, with serious health consequences. In my opinion, the manuscript can be accepted with a minor revision. Please find several suggestions for improving the ms.
We would like to thank the Reviewer for important comments. Referring to suggested topics will increase the scientific value of the manuscript.
Comments
1. The font style sohuld be checked as it changes throughout the text. Please uniform the style.
Ad.1. Suggested unification has been performed.
- Unnecessary highlinings should be removed throughout the text, e.g., line 3, 32, 207, 318, . Please revise the entire text accordingly.
Ad.2. The highlights mentioned by the Reviewer were made by the Editorial Office. The original version of manuscript contained no highlights.
- I suggest using Italic font for Latinisms, only
Ad.3. Appropriate changes has been made (lines 237-240, 266, 311, 370-372, etc.)
- Line 98 this sub-head title should be in bold style
Ad. 4. The mentioned heading was bold in the original manuscript and the change into not bold style must have been made while fitting the manuscript into the journal template by the Editorial Board.
- More literature should be included. Several sentences/paragraphs are lacking in supporting references, e.g., introduction, lines 199-203, lines 243-250, as well as other sentences
Ad. 5. Adequate references has been added to the introduction and to the lines 199-203; in many other paragraphs appropriate references were included as well (over 30 positions more compared to the submitted manuscript). Lines 243-250 (the last paragraph in the introduction of high-risk HPV types), however, has not been supported with more literature references. Our aim was just to list the possible reasons of HPV false negative results here and these have been discusses later on with wide range of references.
- Besides the traditional/clinical cervical cancer screening approaches, novel methods can be, at least briefly, quoted in the manuscript. For instance, the novel and highly reliable High-risk HPV-based droplet digital PCR is acquiring frowing importance in identifying HPV-driven tumors as well as HPV-driven pre-tumoral lesions, including HPV-positive CIN lesions (doi: 10.3389/fmicb.2020.591452 and doi: 10.1007/s13402-017-0331-y). In the future, the ddPCR, might complement current follow-up analyses as well as traditional diagnostic approaches for HPV-driven tumor early identification. This is an important point that, alongside supporting references, should be included.
Ad. 6. Our approach was to revise screening tests and diagnostic methods that have been currently employed into organized or opportunistic cervical cancer screening programmes. In response to the Reviewer’s suggestion we have included a section on prospects in cervical cancer screening where ddPCR and first-void urine sampling were shortly described with appropriate references.
- Lines 138-139 the sentence can be improved in terms of readability
Ad. 7. Appropriate correction has been made.
- In the section 2, a brief mention of the currently employed vaccines would be helpful for the reader.
Ad. 8. A short paragraph regarding prophylactic HPV vaccines has been included in Section 2.
- Line 224 “remaining genotypes” the unnecessary space between words should be removed. The same consideration can be made in lines 489, 551, 556
Ad. 9. Double spaces have been removed throughout the text.
Line 256 “ADC”, should be Adenocarcinoma (ADC), the first time being quoted. The same consideration can be made for Visual Inspection with Acetic Acid (VIA) in line 360
Ad. 10. The first time the word ‘Adenocarcinoma’ occurs in the text is in line 188 and according abbreviation is placed there; similarly, ‘visual inspection with acetic acid’ was quoted for the first time in line 113 as is properly abbreviated there as well.
- Section 3, line 471 additional genes have been described as differentially methylated in cervical cancer (https://doi.org/10.3389/fgene.2020.00347)
Ad. 11. Indeed, the possible connection between methylation of a number of genes and cervical precancer and cancer development has been studied recently as mentioned in the section related to the methylation test. The information about all marketed methylation tests for cervical cancer detection has been added. However, validation and implementation into screening of methylation test of FAM19A4 / hsa-miR124-2 genes seems to be at the most advanced stage at present and therefore remains the only one extensively described in the manuscript.
Conclusions should be a stand-alone section
Ad. 12. An appropriate heading has been added.
